# Augmenting Non-Collaborative Dialog Systems with Explicit Semantic and Strategic Dialog History

**Yiheng Zhou**[♡]   **Yulia Tsvetkov**[♡]   **Alan W Black**[♡]   **Zhou Yu**[◇]

[♡]Language Technologies Institute, Carnegie Mellon University
[◇]Computer Science Department, University of California, Davis
{yihengz1, awb, ytsvetko}@cs.cmu.edu, joyu@ucdavis.edu

## ABSTRACT

We study non-collaborative dialogs, where two agents have a conflict of interest but must strategically communicate to reach an agreement (e.g., negotiation). This setting poses new challenges for modeling dialog history because the dialog's outcome relies not only on the semantic intent, but also on tactics that convey the intent. We propose to model both semantic and tactic history using finite state transducers (FSTs). Unlike RNN, FSTs can explicitly represent dialog history through all the states traversed, facilitating interpretability of dialog structure. We train FSTs on a set of strategies and tactics used in negotiation dialogs. The trained FSTs show plausible tactic structure and can be generalized to other non-collaborative domains (e.g., persuasion). We evaluate the FSTs by incorporating them in an automated negotiating system that attempts to sell products and a persuasion system that persuades people to donate to a charity. Experiments show that explicitly modeling both semantic and tactic history is an effective way to improve both dialog policy planning and generation performance.

## 1 INTRODUCTION

In collaborative dialog settings, agents work together and communicate to reach a common goal (He et al., 2017), such as booking flight and making restaurant reservation. Historically, in collaborative setting, the dialog history and structure is modeled explicitly by tracking semantic content, for example, the set of used slot-value pairs (Bowden et al., 2017; Larionov et al., 2018; Zeigler & Mazor, 1995a;b). Prior work also models dialog history implicitly by using an encoder-decoder model (Sordoni et al., 2015; Shang et al., 2015; Vinyals & Le, 2015; Li et al., 2016; Wen et al., 2015; Yao et al., 2015). Although these techniques show promising results in a collaborative setting, they have drawbacks when applied to non-collaborative settings, where agents have competing interests and goals but aim to reach an agreement, and they use various strategies and tactics to reach an agreement favorable to them. In non-collaborative dialog settings, leveraging effective sequences of tactics is as important as controlling for semantic content, and different tactic sequences lead to different outcomes (Zhou et al., 2019).

Learning latent dialog structure efficiently is challenging for dialog systems. Prior work mainly focused on applying hidden Markov models (HMMs) to capture contextual dependencies within dialogs (Chotimongkol, 2008; Ritter et al., 2010; Zhai & Williams, 2014). Recently, Shi et al. (2019) proposed to use a discrete variational recurrent neural network (D-VRNN) for learning latent dialog structure because of its flexibility and nonlinear nature. In this paper, we take a different approach by using pre-trained FSTs to learn latent dialog structure. FSTs have been used in many traditional dialog systems and have proven to be effective across different domains (Larionov et al., 2018; Zeigler & Mazor, 1995a;b, *inter alia*).

We focus on modeling dialog in non-collaborative settings, and propose to explicitly leverage the dialog structure, including history of tactics and dialog acts, to improve dialog planning and generation. Specifically, we use weighted FSTs to learn dialog acts and tactics history and then integrate the learned FSTs to encoder-decoder pipeline to make the end-to-end system capture semantic and

tactic history. FSTs have several advantages over a traditional recurrent neural network. First, an FST can explicitly track the entire path it traversed, which gives additional symbolic constraints and information about the dialog history. Due to the more informative history representation, an FST has a better prediction of the next step (dialog act/strategy) compared to an RNN, as we empirically confirm. A trained FST serves as a scaffold for dialog history tracking. Second, FST is more interpretable, as each state is explicitly represented by an action distribution. It is thus easier for humans to interpret model decision.

To leverage pre-trained FSTs, we propose an *FST-enhanced hierarchical encoder-decoder model* (FeHED). Our model, depicted in Figure 1, consists of a natural language understanding (NLU) module, two pre-trained FSTs, and a natural language generation (NLG) module. The NLU module has a set of classifiers, where each one is responsible for detecting a dialog act or a negotiation/persuasion strategy/tactics from a given utterance. FSTs model a latent dialog structure, which can encode dialog history. One FST is trained on sequences of dialog acts (FST-DA) and the other FST is trained on sequences of strategies (FST-S). The NLG module is a hierarchical encoder-decoder model (HED), which conditions on the outputs from FST-{DA, S} and previous utterances to predict strategies and generate system response.

We focus on (1) a bargaining scenario where a seller negotiates with a buyer over a given product through a chat interface online (He et al., 2018), and (2) a persuasion dialog setting where a persuader, in an online chat, attempts to persuade the persuadee to donate their earnings to a charity (Wang et al., 2019). We propose an automated agent that plays the role of the seller/persuader. Existing work only focuses on dialog acts, such as "disagree", which capture shallow semantics. However, these dialog acts cannot capture the pragmatics of the dialog acts. For example, whether the user disagrees politely or rudely impacts the dialog system's behavior. To capture pragmatic content, we employ negotiation strategies and tactics introduced by Zhou et al. (2019), motivated by negotiation literature. For persuasion dialog setting, we adopt a set of persuasion strategies from Wang et al. (2019). Besides pragmatics, these strategies also capture domain-specific semantics that dialog acts do not cover.

We evaluate our seller/persuader models using standard measures, BLEU (Papineni et al., 2002) and accuracy of predicting strategies. Additionally, we propose unigram and bigram accuracy of strategies to evaluate dialog models fairly. Experiment results show that FeHED significantly outperforms all the baseline models on four of the five metrics. Moreover, quantitative analysis shows that it is important to model both semantic and tactic history. Finally, qualitative analysis demonstrates that FSTs can track tactic history better than a RNN in non-collaborative settings.

## 2 MODEL

Figure 1 gives an overview of FeHED. There are four components in FeHED: a dialog act classifier, a strategy classifier, two finite state transducers (FST-DA/S), and a hierarchical encoder-decoder model (HED). The four components are connected. The output of the dialog act classifier and strategy classifier is the input of the FSTs. The FSTs' output is the input for HED along with utterance embedding. Finally, HED generates both the next strategy and the next utterance.

1. **Dialog Act Classifier** converts utterances, denoted by $u_1, u_2, .., u_t$, into a sequence of dialog acts $da_1, da_2..., da_t$

2. **Strategy Classifier** converts utterances into a sequence of strategies and tactics $st_1, st_2, ..., st_t$ used in each utterance.

3. **FST-DA/S** takes a sequence of dialog acts $da_1, da_2, ..., da_t$ or strategies $st_1, st_2, ..., st_t$ (green dotted lines in Figure 1) and returns a sequence of state embeddings $s_1^{da/st}, s_2^{da/st}, ..., s_t^{da/st}$.

4. **HED** conditions on $s_1^{da/st}, s_2^{da/st}, ..., s_t^{da/st}$ (indicated by blue dotted lines) and $u_1, u_2, ..., u_t$ to predict a set of possible strategies $st'_{t+1}$ in the next utterance and generate the response $u_{t+1}$.

We describe each component next.

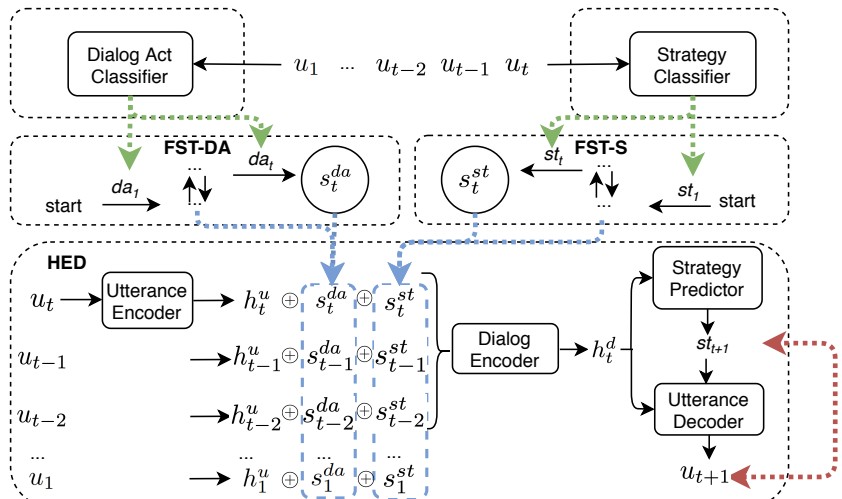

Figure 1: FST-enhanced hierarchical encoder-decoder (FeHED). FeHED first takes dialog history $u_1, u_2, ..., u_t$ and feeds it to dialog act and strategy classifiers. Dialog act/strategy classifiers then output a sequence of dialog acts $da_1, da_2, ..., da_t$ or strategies $st_1, st_2, .., st_t$, which, in turn, are fed to FSTs (green dotted lines). FSTs then output a sequence of state embedding $s_1^{da/st}, s_2^{da/st}, ..., s_t^{da/st}$ to the hierarchical encoder-decoder (HED; blue dotted lines). Lastly, HED generates the system response $u_{t+1}$ and predicts the next strategies $st'_{t+1}$. The red dotted line indicates a posterior constraint. $\oplus$ represents concatenation.

## 2.1 DIALOG ACT CLASSIFIER

We follow the setting in He et al. (2018) and define a list of seven dialog acts, including *introduction, initiate-price, insist on price, agree, disagree, inform and inquire*. He et al. (2018) detect these dialog acts with a set of handcrafted regular expressions. Details about dialog acts are given in Appendix A.1.

## 2.2 STRATEGY CLASSIFIERS

We use fifteen negotiation tactics, proposed by Zhou et al. (2019), operationalizing strategies and tactics described in economics and behavioral science research on negotiation (Pruitt, 1981; Bazerman et al., 2000; Fisher & Ury, 1981; Lax & Sebenius, 2006; Thompson et al., 2010). These include rhetoric strategies, e.g., *talk informally, use certainty words*, and behavioral strategies, e.g., *build rapport, address buyer's concerns*. Part of these strategies are implemented using regular expressions, for example, "please" is a salient word for *communicate politely*. Other strategies are implemented using linear classifiers, trained on negotiation dialogs. Details about negotiation strategies are given in Appendix A.2.

We adopt a set of persuasion strategies, developed by Wang et al. (2019), based on the Elaboration Likelihood Model (ELM) theory from social psychology (Petty & Cacioppo, 1986). Persuasion strategies include persuasive appeal, e.g., *logical appeal, emotion appeal,* and persuasive inquiry, e.g., *source-related inquiry, personal-related inquiry*. These strategies are captured with hybrid recurrent convolutional neural network classifiers (Wang et al., 2019). Details about persuasion strategies are given in Appendix A.3.

## 2.3 FSTs FOR SEMANTIC AND STRATEGIC DIALOG HISTORY

Two FSTs were trained to learn latent dialog structure. One is trained with dialog act sequences, while the other one is trained with strategy sequences.

Training data for FSTs is sequences of dialog acts or strategies, where each sequence is extracted from a dialog by using dialog act/strategy classifier. We initialize FST with a single ergodic node

where all edges go back to itself. We then iterate over this initial FST and split the state(s). We greedily select a state split, based on splitting on the incoming dialog act/strategy that minimizes the entropy of the two generated states. This follows the algorithm often used in node splitting in building decision trees (Breiman et al., 1984). There is no well-defined stopping criteria, therefore, we need to choose the number of states as a hyperparameter. We show an example of trained FST-DA with 3 states in Appendix A.4. By looking at incoming and outgoing edges of a node, we can understand which dialog state this node represents. For example, if the incoming edge of a state is "seller answering a question" and the outgoing edge is "buyer inquiring", then in this state, the buyer is most likely to ask a question.

The trained FST gives a probability mass function (PMF) for the likelihood of transfer from the current state to each of the possible dialog acts. We use this vector of probabilities as a state embedding $(s_t^{da/st})$, which can be generated for each utterance and concatenated to other utterance representations. At test time, we use our dialog act or strategy classifiers to choose the predicted dialog act or strategy, and transition to the new state in the FST to get the PDF for the next set of dialog acts. The FST not only returns the current state embedding, but also all the embeddings of state it traversed since the start state.

## 2.4 HIERARCHICAL ENCODER-DECODER(HED)

Let $u_t = [w_1^t, ..., w_n^t]$, where $w_i^t$ is the $i$-th word in current utterance $u_t$. We use a standard GRU (Cho et al., 2014) to encode $u_t$ into a hidden state representation $h_t^u$.

$$h_t^u = \text{GRU}^u(u_t)$$

We concatenate the utterance embedding with the output of FST-DA and FST-S to enrich the utterance embedding to incorporate dialog history. $h_t^{\prime u} = [h_t^u; s_t^{da}; s_t^{st}]$. Finally, we use another GRU to combine all utterances till current time to encode the entire dialog into a hidden state $h_t^d$

$$h_t^d = \text{GRU}^d\left(h_1^{\prime u}, h_2^{\prime u}, ..., h_t^{\prime u}\right)$$

Then we predict the next utterance strategies $st_{t+1}$ and finally, generate system response $u_{t+1}$ using $h_t^d$.

**Strategy predictor**  Before generating system response, we add an intermediate step to predict the set of possible strategies $st_{t+1}$ in it. The output $st_{t+1}$ is a 15-dimensional binary-value vector, where each dimension represents whether a certain negotiation strategy occurred in $u_{t+1}$. We compute the probability of the $j$-th strategy occurring in $u_{t+1}$ as:

$$p(st_{t+1,j} = 1 | h_t^d) = \sigma(W_j[h_t^d] + b_j)$$

where $W_j$ and $b_j$ are learned parameters. We denote the negative log likelihood of strategies $\mathcal{L}_{ST}$ as the loss function of this task:

$$\mathcal{L}_{ST} = - \sum_{\{j | st'_{t+1,j}=1\}} log(st_{t+1,j}) - \sum_{\{j | st'_{t+1,j}=0\}} log(1 - st_{t+1,j}),$$

where $st'_{t+1}$ is the ground truth strategies.

**Utterance decoder** is a standard GRU decoder with attention (Bahdanau et al., 2015). The input of this decoder is the dialog hidden state $h_t^d$ concatenated with the predicted strategies $st_{t+1}$. It calculates a probability distribution $p_j^{t+1}$ over vocabulary at time $j$ conditioning on the previous word:

$$p_j^{t+1} = \text{softmax}(\text{GRU}^{de}([st_{t+1}; h_t^d], w_{j-1}^{t+1}))$$

Cross entropy loss $\mathcal{L}_{NLG}$ is used for this generation task:

$$\mathcal{L}_{NLG} = - \sum_{\{w_j \in u'_{t+1}\}} log(p_{j,w_j}^{t+1}),$$

where $u'_{t+1}$ is the target utterance.

Finally, we combine the strategy prediction task and system utterance generation loss together. We also add a posterior constraint to enforce the generated utterances and the predicted negotiation strategies align with each other:

$$\mathcal{L}_{joint} = \mathcal{L}_{NLG} + \alpha\mathcal{L}_{ST} + \beta\sum_j \mathbb{1}(st_{t+1,j} \notin u_{t+1})$$

where $\alpha, \beta$ are constants (both were set to 1.0 in our experiments) and the last term is a posterior constraint (the red dotted line in Figure 1) that has a positive value if the strategies in the generated utterance $u_{t+1}$ does not contain some strategies in the predicted strategies $st_{t+1}$. We jointly train strategy predictor and utterance decoder using $\mathcal{L}_{joint}$.

## 3 EXPERIMENTS

**Datasets**    We evaluate our model's performance on two non-collaborative dialog data sets, CraigslistBargain (He et al., 2018) and Persuasion For Good (Wang et al., 2019). CraigslistBargain consists of dialogs of two people buying and selling products. The negotiation scenarios are crawled from `craigslist.com`, which includes a product description, an optional product photos, and its listing price. The buyer is given a private target price which is strictly lower than the listing price. The data was collected on Amazon Mechanical Turk (AMT) platform with two Turkers role-playing with each other. The seller aims to obtain as much profit as possible while the buyer tries to purchase the product with a price close to the private target price. Both parties are encouraged to reach an agreement. There are in total 5,383 dialogs for training, 643 for validation and 656 for testing. The average conversation length is 9.2 turns. The vocabulary size of the data is 13,928.

Persuasion For Good dataset is also collected on AMT, where two workers were randomly assigned the roles of persuader and persuadee. The goal of the persuader is to persuade the persuadee to donate his/her task earning to a charity, Save the Children. There are 1,017 dialogs, where 300 are annotated with persuasion strategies. We split the dataset into 180 training dialogs, 60 for validation and 60 for test. The average conversation length is 10.43 turns. The vocabulary size is 8,141.

**Experimental setup**    We train each model for 20 epochs and choose the one that performs best on validation dataset. We use a mini-batch of 20 and learning rate of $5e^{-4}$ for encoders and $5e^{-3}$ for utterance decoder. The encoders and decoder are GRUs, each has two layers and a hidden state size of 300.

We compare FeHED to a list of baselines and present their results in Table 1.

- **HED**: A vanilla HED that does not consider dialog act nor strategy.
- **FeHED+CE**: We replace the indicator in $\mathcal{L}_{joint}$ with cross-entropy loss.
- **FeHED−SP**: To test the importance of modeling strategies alone, we remove everything that involves strategies, specifically, strategy prediction, tracking and decoding.
- **FeHED−FSTs**: To test the importance of incorporating semantic and strategic history, we remove both FST-DA and FST-S from FeHED.
- **FeHED−FST-S/DA**: To test which type of history is more important, semantic or tactic, we remove either FST-S or FST-DA from FeHED.
- **HED+RNN**: We compare our approach with He et al. (2018)'s method, by replacing FST in FeHED with an RNN encoding tactic sequence.
- **Sequicity**: Lei et al. (2018) use *belief span* to model dialog state and improve system performance. However, Sequicity cannot be directly applied to our problem. For a fair comparison, we replace Lei et al. (2018)'s slot-values in a belief span with dialog acts and strategies.

**Evaluation**    We evaluate FeHED's performance (1) on the ability of generating high quality responses, and (2) on whether the responses carry the correct strategy. We also explore the effectiveness of history tracking by performing an ablation study. Lastly, we conduct human evaluation to test the dialog system's persuasiveness, coherence and naturalness.

To evaluate FeHED's responses we use four types of metrics:

- Strategy predictor's F1 and accuracy (**S.F1** and **S.acc**). We evaluate strategy prediction performance along with response generation quality, to assess strategy tracking. However, not every model in Table 1 can output strategy. For these models, we replace their utterance decoder with a strategy predictor and retrain the model by optimizing $\mathcal{L}_{ST}$. We evaluate the retrained model by measuring its F1 score and accuracy.
- We use **BLEU** to evaluate generation quality. Although in other generation tasks BLEU is a standard measure, in dialog modeling it is not a reliable metric, as different sentences can convey the same content in different ways.
- Utterance decoder's accuracy of generated strategies (**Uni.acc**, **Bi.acc**). We first apply our strategy classifier to extract strategies in generated utterance. Then, the extracted strategies are compared with ground truth to calculate the accuracy (Uni.acc). Due to the nature of dialogs, multiple strategies can be appropriate given the same dialog history. Therefore, we expand the ground truth strategy set by sampling dialogs with similar dialog history (previous two turns). We use the expanded ground truth set to calculate bigram accuracy (Bi.acc).
- **Human evaluation**.[1] We also conducted two types of human evaluation for the negotiation task: (1) third-person rating; (2) second-person rating. For third-person rating, we randomly give each participant four different types of dialogs and ask him/her to rate each dialog's seller in terms of persuasiveness, coherence, and naturalness (on a scale of 1 to 5). These dialogs have FeHED, FeHED−FST-DA, FeHED−FST-S or HED to play the role of seller. For second-person rating, we ask participants to conduct four conversations by playing the role of buyer to negotiate with FeHED, FeHED−FST-DA, FeHED−FST-S, and HED, respectively. Then, we ask them to compare and rate each model in terms of persuasiveness, coherence, and naturalness (on a scale of 1 to 5).

## 3.1 RESULTS

| | Negotiation | | | | | Persuasion | | | | |
|---|---|---|---|---|---|---|---|---|---|---|
| **Models** | Uni.acc | Bi.acc | S.acc | S.F1 | BLEU | Uni.acc | Bi.acc | S.acc | S.F1 | BLEU |
| FeHED | 49.6 | 59.3 | **61.9** | **22.8** | 20.6 | **0.18** | 0.77 | **0.80** | **18.2** | 13.5 |
| FeHED+CE | **50.3** | **59.6** | 60.4 | 21.5 | 20.7 | 0.18 | **0.78** | 0.79 | 18.2 | 13.7 |
| −FSTs | 42.3 | 55.1 | 42.4 | 19.5 | 20.5 | 0.13 | 0.70 | 0.80 | 15.6 | 13.6 |
| −FST-S | 43.1 | 54.3 | 46.7 | 20.2 | **20.9** | 0.12 | 0.70 | 0.75 | 17.4 | 13.8 |
| −FST-DA | 42.8 | 54.9 | 49.2 | 20.8 | 20.3 | 0.10 | 0.70 | 0.80 | 16.8 | 14.0 |
| −SP | 46.5 | 56.3 | 47.3 | 20.4 | 20.5 | 0.15 | 0.67 | 0.78 | 18.0 | 13.9 |
| HED+RNN | 46.5 | 56.8 | 57.2 | 15.5 | 20.3 | 0.16 | 0.75 | 0.77 | 17.9 | 13.6 |
| Sequicity | 44.0 | 57.9 | - | - | 16.2 | - | - | - | - | - |
| HED | 36.9 | 51.2 | 38.4 | 15.6 | 20.8 | 0.12 | 0.66 | 0.77 | 15.8 | **14.1** |

Table 1: Ablation and baseline results. FeHED achieves the best performance on all metrics except BLEU. Moreover, removing any component results in a significant decrease in performance.

Table 1 shows the result of negotiation and persuasion dialogs. FeHED achieves the best performance on all metrics except BLEU. However, single-reference BLEU assumes only one possible system response while dialog system can have multiple valid responses. Comparing with a vanilla HED model, FeHED improves S.acc by +23.5, S.F1 by +7.2, Uni.acc by +12.7 and Bi.acc by +8.1, while maintaining comparable BLEU score. This suggests that incorporating semantic and tactic information leads to a better dialog system. We also evaluate model performance when ablating one or more model components. Result shows that removing any FST causes a worse performance, which suggests modeling both semantic and strategic history is necessary. Moreover, the HED+RNN setup confirms that FSTs better model semantic and tactic history than RNNs in non-collaborative settings. Noticeably, all models' S.F1 are low, which may be due to the fact that the negotiation strategy set is large and some of them have low frequency in training data, therefore causing overall low performance. We also observe that S.acc is higher than Uni.acc for all the models. This is expected, because predicting negotiation strategies is more straightforward than generating system utterances with the correct negotiation strategies. Although Sequicity has a very high Uni.acc and Bi.acc, it has a much lower BLEU score compared to all the other models except FeHED. It is likely because

---

[1]The study was approved by the IRB.

Sequicity uses CopyNet (Gu et al., 2016), which is not designed for non-collaborative dialogs but rather collaborative tasks (e.g. booking a restaurant). Negotiation and persuasion tasks yield similar results, confirming that the benefit of using FSTs to model both semantic and strategic history is not limited to a single domain.

| Models | Second-person Rating | | | | Third-person Rating | | | |
| --- | --- | --- | --- | --- | --- | --- | --- | --- |
| | Persuasive | Coherent | Natural | Sale Price | Persuasive | Coherent | Natural | Sale Price |
| FeHED | **2.6** | **2.8** | **3.0** | **0.84** | **3.2** | **3.9** | 3.5 | **0.68** |
| −FST-DA | 2.5 | 2.2 | 2.4 | 0.70 | 3.0 | 3.4 | 3.5 | 0.64 |
| −FST-S | 2.0 | 2.4 | 2.4 | 0.64 | 2.9 | 3.4 | 3.3 | 0.59 |
| HED+RNN | 2.3 | 2.5 | 2.6 | 0.49 | 2.8 | 3.8 | **3.6** | 0.44 |
| HED | 1.8 | 1.9 | 1.9 | 0.62 | 2.9 | 3.4 | 3.1 | 0.50 |

Table 2: Human evaluation ratings for (on a scale from 1 to 5) for FeHED, FeHED−FST-DA, FeHED−FST-S, and HED. We conducted third-person rating and second-person rating. Sale price is normalized.

Table 2 shows the results of human evaluation. For third-person rating, we asked an expert to generate 20 dialogs by negotiating with FeHED, FeHED−FST-DA, FedHED−S, HED+RNN and HED respectively (5 dialogs each). We then recruited 50 people on AMT to rate these generated dialogs. Result shows that FeHED is more persuasive, coherent and natural than all the baselines. For second-person rating, we asked 50 participants on AMT to play the role of buyer to negotiate with FeHED, FeHED−FST-DA, FeHED−FST-S, HED+RNN and HED respectively. Results show that FeHED outperforms RNN and other models on all the metrics except naturalness. It is likely because RNN is trained jointly with HED, but it is not good at explicitly tracking and preserving the history. Although FST is learned separately, it forces the model to learn the history through a list of traversed states. We analyze the generated dialogs for human evaluation and find that baselines are more likely to accept unfair offers and apply inappropriate strategies.

---

**Listing Price:** 40
**Buyer's Target Price:** 36
**Title:** 2017 NEW Stans 24 and 26 Tubeless Tire Kit
**Product Description**: "I have a NEW Stan's Tubeless tire Conversion kit for sale for 24 AND 26 wheels/tires. Pick up in Walnut Creek, CA in evenings"

| Seller (FeHED): S Buyer (Human): B | Seller (HED): S Buyer (Human): B |
| --- | --- |
| B: Hello! | B: Hello! |
| S: Hi | S: Hello |
| FST: ⟨*intro, 0.49*⟩, ⟨*init-price, 0.19*⟩... | |
| B: How old is your kit? | B: How old is your kit? |
| S: I only had it for months. | S: It's only a year old. |
| FST: ⟨*inform, 0.51*⟩, ⟨*disagree, 0.20*⟩... | |
| B: The price is a bit high, can you do 30? | B: The price is a bit high, can you do 30? |
| S: **I can't go that low. I could go down to 38.4.** | S: **I can do that.** |
| FST: ⟨*propose price, 0.51*⟩, ⟨*agree, 0.11*⟩... | |
| B: How about 35? | S: ‹offer 30› |
| ... | B: ‹accept› |

| Seller (FeHED): S Buyer (Human): B | Seller (HED+RNN): S Buyer (Human): B |
| --- | --- |
| B: Hello! | B: Hello! |
| S: Hello | S: Hello |
| B: How much are you looking for? | B: How much are you looking for? |
| S: I'm asking 40 for it. | S: I'm asking 40 for it. |
| B: Can you do 30 dollars and I can pick it up? | B: Can you do 30 dollars and I can pick it up? |
| S: How about 37.6 ? **I will deliver to you.** | S: Can you do 37.6? |
| FST: ⟨*negotiate side offers, 0.49*⟩ | ... |
| ... | |

Table 3: Examples of generated responses with and without FSTs.

Finally, we show and example of a negotiation dialog. The first example in Table 3 shows the dialog with the bot that uses history tracking (FeHED). For the examples with FeHED, we also show top most likely dialog acts given by the current state of FSTs. As shown in bold text in the first two examples in Table 3, HED tends to agree with buyer's proposal although they have just started negotiating the price. On the other hand, FeHED adopts a more appropriate action: rejects buyer's proposal and gives a counter price. This is likely because FSTs give the system a state embedding, where ⟨*propose price*⟩ has the highest probability (0.51), which tells the system that in the current state, the probability of making a counter price is 0.51. Besides ⟨*propose price, 0.51*⟩, other generated utterances are mostly following the top most likely dialog acts proposed by FSTs (e.g. ⟨*intro, 0.49*⟩, ⟨*inform, 0.51*⟩).

Table 1 shows that FST models dialog history better than RNN in non-collaborative setting. Two examples in the last row of Table 3 provide an example. Noticeably, FeHED recognizes a tactics used in the previous utterance ("pick it up") and responds "I will deliver to you"; but with RNN, the bot ignores the previous tactic history and proposes another price. Presumably, this is because FST explicitly captures tactics used in the history, while RNN does not.

## 4  RELATED WORK

Our work extends the line of research on non-collaborative dialog tasks, such as negotiation and persuasion. Lewis et al. (2017) demonstrated a task where a collection of items are divided between two agents. Some prior work also focus on a strategic board game called Settlers of Catan where players can offer resources in exchange for others and they can also reply to offers from other players (Cuayáhuitl et al., 2015; Keizer et al., 2017). However, these tasks do not require modeling rich communication skills, but focus on decision-making skills. Therefore, prior studies on these tasks only focus on tactic history but ignore semantic content. To consider both semantic and tactic history, we choose a bargaining scenario proposed by He et al. (2018), where a seller and a buyer negotiate price over an item for sale. To show the generalizability of our work, we also choose a persuasion dialog setting proposed by Wang et al. (2019), where a persuader tries to strategically convince a persuadee to donate his/her earnings to a charity.

Most end-to-end approaches incorporate history through hidden states (Sordoni et al., 2015; Shang et al., 2015; Vinyals & Le, 2015; Li et al., 2016; Wen et al., 2015; Yao et al., 2015). Such methods only focus on capturing semantic history. Lei et al. (2018) proposed a text span called *belief span* for encoding dialog history, which is combined with a simple seq2seq model to improve dialog generation. Specifically, belief span tracks entities mentioned so far (e.g. restaurant types) to explicitly model dialog history. We utilize trained FSTs to encode dialog history instead of a text span. Additionally, it requires human annotations as supervision to train a belief span, while our FSTs are fully unsupervised. Therefore, our FSTs can be applied to other domains easily.

Rule-based dialog modules incorporate history using symbolic rules. Larionov et al. (2018); Zeigler & Mazor (1995a;b) use a finite-state automata to keep track of dialog context. Fang et al. (2018) suggests building a hierarchical dialog manager that keeps track of engagement, coherence, and user experience. Bowden et al. (2017) utilizes a state graph structure to model dialog flows. He et al. (2018) applies a neural model to predict a sequence of dialog acts as dialog manager. We also utilize finite-state machine in FeHED, but it is automatically learned using unsupervised data. Moreover, we use FSTs to learn the dialog structure instead of using it directly as the dialog manager.

## 5  CONCLUSION

In non-collaborative dialog settings, it is important to not only track the semantic intent, but also strategies and tactics that express this intent. To improve non-collaborative dialog planning and generation, we propose to explicitly model both semantic and tactic history by using automatically trained FSTs. We then evaluate the trained FSTs on a negotiation dialog system and a persuasion dialog system. Result shows that explicitly modeling both semantic and tactic history achieves the best performance. We have also shown that FST models tactic history better than a RNN in non-collaborative dialog settings.[2]

---

[2]All sources and data will be publicly released.

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

# A  APPENDIX

## A.1  DIALOG ACTS

| Meaning | Dialog Act | Example | Detector |
|---|---|---|---|
| Greetings | ⟨*intro*⟩ | *"Hello there!"* | rule |
| Propose the first price | ⟨*init-price*⟩ | *"Can you do 30 dollars?"* | rule |
| Insists on an offer | ⟨*insist*⟩ | *"I can't go lower than 30 dollars."* | rule |
| Agree with the current proposal | ⟨*agree*⟩ | *"Ok, you have a deal."* | rule |
| Disagree with the current proposal | ⟨*disagree*⟩ | *"sorry I can't go that low."* | rule |
| Answer buyer's question | ⟨*inform*⟩ | *"This bike is brand new."* | rule |
| Ask a question | ⟨*inquire*⟩ | *"Which color do you prefer?"* | rule |

Table 4: A list of dialog acts from (He et al., 2018).

## A.2 NEGOTIATION STRATEGIES

| Negotiation Strategy | Action | Example | Detector |
|---|---|---|---|
| Focus on interests, not positions | ⟨*Describe product*⟩ | "*The car has leather seats.*" | classifier |
| | ⟨*Rephrase product*⟩ | "*45k miles*" → "*less than 50k miles*" | classifier |
| | ⟨*Embellish product*⟩ | "*a luxury car with attractive leather seats*" | classifier |
| | ⟨*Address concerns*⟩ | "*I've just taken it to maintainence.*" | classifier |
| | ⟨*Communicate interests*⟩ | "*I'd like to sell it asap.*" | classifier |
| Invent options for mutual gain | ⟨*Propose price*⟩ | "*How about $9k?*" | classifier |
| | ⟨*Do not propose first*⟩ | n/a | rule |
| | ⟨*Negotiate side offers*⟩ | "*I can deliver it for you*" | rule |
| | ⟨*Hedge*⟩ | "*I **could** come down a bit.*" | rule |
| Build trust | ⟨*Communicate politely*⟩ | greetings, gratitude, apology, "*please*" | rule |
| | ⟨*Build rapport*⟩ | "*My kid really liked this bike, but he outgrew it.*" | rule |
| | ⟨*Talk informally*⟩ | "*Absolutely, ask away!*" | rule |
| Insist on your position | ⟨*Show dominance*⟩ | "*The absolute highest I can do is 640.0.*" | rule |
| | ⟨*Negative sentiment*⟩ | "*Sadly I simply cannot go under 500 dollars.*" | rule |
| | ⟨*Certainty words*⟩ | "*It has **always** had a screen protector*" | rule |

Table 5: A list of negotiation strategies from Zhou et al. (2019).

## A.3 PERSUASION STRATEGIES

| Persuasion Strategy | Action | Example | Detector |
|---|---|---|---|
| Logical appeal | ⟨*Use reasoning*⟩ | "*This donation will make an impact for children.*" | classifier |
| Emotion appeal | ⟨*Elicit specific emotions*⟩ | "*Millions of children are facing violence.*" | classifier |
| Credibility appeal | ⟨*Cite organizational impacts*⟩ | "*This charity has a program called sponsor child.*" | classifier |
| Foot-in-the-door | ⟨*Start with small donation*⟩ | "*How about we donate 0.2 each first ?*" | classifier |
| Self-modeling | ⟨*Make donation myself*⟩ | "*I want to donate some amount from this survey.*" | classifier |
| Personal story | ⟨*Tell personal donation story*⟩ | "*I donated 1 dollar to this charity before.*" | classifier |
| Donation information | ⟨*Give information of donation*⟩ | "*Research team will send money to this charity.*" | classifier |
| Source-related inquiry | ⟨*Ask about the charity*⟩ | "*Have you heard of Save the Children before?*" | classifier |
| Task-related inquiry | ⟨*Ask opinion of donation*⟩ | "*Are you interested in donating some money?*" | classifier |
| Personal-related inquiry | ⟨*Ask personal experience*⟩ | "*Have you ever donated to any charity before?*" | classifier |

Table 6: A set of persuasion strategies from Wang et al. (2019).

## A.4 DIALOG STATES FST EXAMPLE

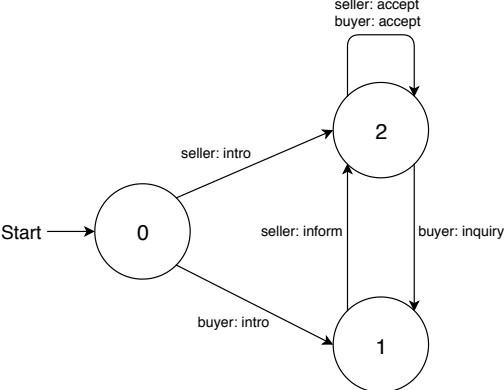

Figure 2: An example of FST-DA with three states. For each edge, we only show the top frequent dialog acts for better visualization.

