# OpenReview forum: "Augmenting Non-Collaborative Dialog Systems with Explicit Semantic and Strategic Dialog History"
_ICLR.cc/2020/Conference — Accept (Poster)_

### Official Review · AnonReviewer1 · 2019-10-22
**Official Blind Review #1**

**Rating:** 6

**Review:**

This paper proposed a framework for non-collaborative dialog systems. To track the history better, the authors propose to apply two pre-trained finite state transducers (FSTs) to take the sequence of dialog acts or strategies as inputs and output sequences of state embeddings for the hierarchical encoder-decoder (HED) part as inputs. The authors test their model on two tasks, a bargain task and a persuasion task.

The model that the authors prepose makes sense to me. It seems that the FSTs can successfully help the model to track the history better and propose useful state embeddings for the rest part of the model, on both dialog acts and strategies. Also, the authors have done a comprehensive empirical study to show the gain of the proposed techniques over the pure HED model and some baseline method. So I think this is a good work. But potentially it would be better if the authors can show more insights and details for the FSTs.

Some detailed comments:

1. The example dialogs in Table 3 show us that the FSTs can output dialog act distributions that can help the agent behave better compared to the case with an RNN for tracking the history, and the case without FST/RNN. This shows the success of the FST framework.

2. The authors have done a comprehensive comparison for the cases with / without the two FSTs, as well as the case without the strategy predictor. Also, the authors compared with another existing method (Sequicity). The proposed method performs well compared to the baseline methods, on both general performances and the two types of human-evaluations. This shows the good empirical performance of the proposed method.

3. This paper can be better if the authors can provide more insights and details for the FSTs. For example, why FST is better than RNN? The authors mentioned something about that, but it is not clear enough and it will be great if the authors can provide more discussions on this. In addition, the output alphabets and the sets of states are not clear for the FSTs, it will be better if the authors show the full details for the FSTs, as in my opinion this is the major contribution of this work.

4. A minor comment: There are some inaccurate expressions in this paper. For example, the authors mentioned that the FSTs about put probabilistic density functions. However, to my understanding, the set of dialog acts is finite, hence FSTs output probabilistic mass functions, not probabilistic density functions. Please proofread about that.

Question:

1. Can you explain more about the choice for the parameters? For example, why the possible strategy output is a 15-dimensional binary-value vector?

2. For the loss L_{joint}, shouldn't it be (st_{t+1,j}\not\in u_{t+1}) in the indicator function?



**Experience Assessment:**

I do not know much about this area.

**Review Assessment: Checking Correctness Of Derivations And Theory:**

N/A

**Review Assessment: Checking Correctness Of Experiments:**

I assessed the sensibility of the experiments.

**Review Assessment: Thoroughness In Paper Reading:**

I read the paper at least twice and used my best judgement in assessing the paper.

---

> ### Author Response · Authors · 2019-11-11
> **Response to AnonReviewer1**
>
> Thank you for your time and thoughtful review!
>
> “Can you explain more about the choice for the parameters? For example, why the possible strategy output is a 15-dimensional binary-value vector?”
>     * We draw insights from behavioral research and operationalize these in 15 negotiation strategies in our strategy prediction task. This is why the strategy output is a 15-dimensional vector.
>
> “For the loss L_{joint}, shouldn't it be (st_{t+1,j}\not\in u_{t+1}) in the indicator function?”
>     * Thank you for catching this, we will fix it in the final version.
>
> “Why FST is better than RNN?”
>     * Empirically, we find that incorporating FSTs to represent the history of dialog acts and the history of negotiation strategies yielded better results than with LSTMs. We hypothesize that this is because RNNs encode dialog history as hidden states, whereas FSTs represents dialog history explicitly as a path through different states, it preserves the history better and more consistently.
>     * We will provide a formal description of FSTs in the final version of the paper, thanks for pointing out the missing details. In summary, at each time step, FST takes a seller/buyer’s action and outputs a probability distribution of a buyer/seller’s next action.
>
> “The set of dialog acts is finite, hence FSTs output probabilistic mass functions, not probabilistic density functions”
>     * Thank you, we will fix it in the final version.

---

### Official Review · AnonReviewer3 · 2019-10-23
**Official Blind Review #3**

**Rating:** 3

**Review:**

This work presents two FST models to explicitly incorporate semantic and strategic/tactic information into dialog systems. Experiments on two datasets from prior work show the advantage of this model.

Given the writing of this paper, I am not sure I understand the novelty of this work on the modeling side. Particularly, section 2.3 is too brief to provide enough technical details so readers can understand how exactly the two FST models work and why they are effective in encoding semantic and strategic information. With the important details being left out, it is difficult to evaluate the novelty of this work.

In addition, although the evaluation in section 3 shows the proposed model outperforms some existing competitive systems. The experiment setup is questionable:

- Both competitive systems (He et al., 2018) and (Lei et al., 2018) are not exactly the same as proposed in these two works. To make a fair comparison, I strongly suggest running the systems released from their works
- The evaluation measurements used in the human evaluation is not directly comparable to prior work, which makes it further unclear about the performance of the proposed model.


Some additional comments on details

- In the paragraph before section 2.4, I think "probability density function" should be "probability distribution", since the random variables are discrete.
- In the joint loss function (the equation before section 3), why use an indicator function instead of the cross-entropy loss for the last term?
- In experimental setup, my understanding is that the model was jointly trained with all components. Why there are two separate learning rates?
- In evaluation, what are the exact definitions of Uni.acc and Bi.acc? I read this paragraph multiple times, and I am still not sure whether I understand it correctly.

**Experience Assessment:**

I have published in this field for several years.

**Review Assessment: Checking Correctness Of Derivations And Theory:**

I carefully checked the derivations and theory.

**Review Assessment: Checking Correctness Of Experiments:**

I carefully checked the experiments.

**Review Assessment: Thoroughness In Paper Reading:**

I read the paper thoroughly.

---

> ### Author Response · Authors · 2019-11-11
> **Response to AnonReviewer3**
>
> Thank you for your helpful reviews! Here are our responses to your concerns.
>
> “Both competitive systems (He et al., 2018) and (Lei et al., 2018) are not exactly the same as proposed in these two works.”
>     * For Lei et al., 2018, we re-used Lei et al.’s implementation with minimum modifications; the original code is not applicable to our task. They use belief span to track entities mentioned so far, while we are tracking dialog acts and negotiation strategies. Therefore, the only modification we performed was to replace the entities in their belief span with our dialog acts and negotiation strategies.
>     * For He et al., 2018, the RNN structure used to model dialog acts is the exact model used by He et al. We implemented two RNNs to model dialog acts and negotiation strategies, because we hypothesize that tracking both dialog acts and negotiation strategies together will lead to better dialog performance.
>
> “The evaluation measurements used in the human evaluation is not directly comparable to prior work.”
>     * Our human evaluation covers more aspects than He et al., 2018’s human evaluation. He et al., 2018 only measures human likeness (corresponding to naturalness in our paper), while we measure in total three aspects: naturalness, coherence and persuasiveness.
>
> "Probability density function" should be "probability distribution"
>     * Yes, you are right. It should be “probability mass function” (as Reviewer #1 pointed out) or “probability distribution”. Thank you for pointing this out, we will fix this.
>
> “In the joint loss function (the equation before section 3), why use an indicator function instead of the cross-entropy loss for the last term?”
>     * Following the Reviewer's suggestion, we reran the experiment with the cross-entropy loss for our best model (FeHED+CE). The table below shows the result. We observe an improvement of generation quality (Bi.acc, Uni.acc, BLEU), but a worse performance for strategy prediction (S.acc, S.F1). It is likely because cross entropy loss puts a constraint on generation quality, which could sacrifice the performance of strategy prediction. Overall, the new model still beats all the other baseline models (except for BLEU score).
>
>                                                 Negotiation
> Models               Uni.acc   Bi.acc     S.acc    S.F1     BLEU
> -----------------------------------------------------------------------------
> FeHED+CE               50.3      59.6       60.4    21.5      20.7
> FeHED                      49.6      59.3       61.9    22.8      20.6
>
>                                                 Persuasion
> Models               Uni.acc   Bi.acc     S.acc    S.F1     BLEU
> -----------------------------------------------------------------------------
> FeHED+CE               0.18      0.78       0.79    18.2      13.7
> FeHED                      0.18      0.77       0.80    18.2      13.5
>
>
> “In experimental setup, my understanding is that the model was jointly trained with all components. Why there are two separate learning rates?”
>     * Sorry for the confusion. The learning rates should be 1.0 for both terms.
>
> “In evaluation, what are the exact definitions of Uni.acc and Bi.acc? I read this paragraph multiple times, and I am still not sure whether I understand it correctly.”
>     * What is Uni.acc and why do we need it? Previous work evaluates the generation quality only using BLEU. BLEU evaluates how close is the generated utterance to the ground truth. However, BLEU is unable to tell us whether the generated utterance uses the correct negotiation strategies, which is very important for non-collaborative dialog. Both BLEU and Uni.acc are used to evaluate the generation quality. However, instead of focusing on the surface form as BLEU does, Uni.acc quantifies the percentage of generated utterances that contain the correct negotiation strategies.
>     * What is Bi.acc and why do we need it? It is not enough to compare with the ground truth utterance in dialog generation. Because in dialogs, multiple responses can be correct given the context. So the ground truth utterance is only one of the many valid responses. To evaluate our model more accurately, we not only calculate a single Uni.acc using the ground truth, but also consider other possible correct strategies given the conversation context using Bi.acc.
>     * Also we want to emphasize all the automatic evaluation metrics are insufficient to evaluate dialog qualities. They only capture partial information of dialog quality. We thus conduct human evaluation and recommend referring to human evaluation as the most comprehensive evaluation.

---

### Official Review · AnonReviewer2 · 2019-10-27
**Official Blind Review #2**

**Rating:** 6

**Review:**

This is an interesting paper about use of Finite State Transducers to model semantic and tactic history in dialogues. This can be fruitful in settings where negotiation settings.

I have some basic questions.

Why not a direct comparison with the models proposed by Wang et al., 2019, He et al., 2019?

Why is human evaluation performed only w.r.t. HED and its variant but not Sequicity?

 How do you come up with number 37.6 in negotiation dialogue?

I think, there is a long way to go to solve the considered dialogue modeling problem. I like the application of FSTs for the problem. Why not compare w.r.t. HMMs as you mention it as a choice in the intro?




**Experience Assessment:**

I have published one or two papers in this area.

**Review Assessment: Checking Correctness Of Derivations And Theory:**

I carefully checked the derivations and theory.

**Review Assessment: Checking Correctness Of Experiments:**

I carefully checked the experiments.

**Review Assessment: Thoroughness In Paper Reading:**

I read the paper thoroughly.

---

> ### Author Response · Authors · 2019-11-11
> **Response to AnonReviewer2**
>
> Thank you very much for your helpful feedback! Here are our responses to your questions
>
> “Why not a direct comparison with the models proposed by Wang et al., 2019, He et al., 2019?”
>      * Wang et al. 2019 only proposed a persuasion dataset and did a classification task to classify persuasion strategies. They did not build a dialog system that we can compare against.
>      *Our HED+RNN in Table 1 and Table 2 is exactly He et al., 2019 ’s method, except that we use two RNNs to model both dialog acts and negotiation strategies, while He et al., 2019 only uses one RNN to modal dialog acts. Moreover, we use the same set of dialog acts as He et al., 2019. Sorry we did not make it clear in the paper. We will update that in the final draft.
>
> “Why is human evaluation performed only w.r.t. HED and its variant but not Sequicity?”
>     * For the automatic metric, Sequicity performs much worse than other baseline models. So we did not have it in human evaluation. We only chose the baselines that has overall best automatic evaluation results to compare against in human evaluation.
>
> “How do you come up with number 37.6 in negotiation dialogue?”
>     * For the negotiation task, a buyer/seller can propose a number with different precision as price, such as 3.444. In text generation, as we need a fixed-size vocabulary, such variance in number is hard to deal with. So, we normalized the price with respect to its original sale price to deal with it. We will clarify this in the final draft.
>
> “Why not compare w.r.t. HMMs as you mention it as a choice in the intro?”
>     * We agree that implementing dialog history with HMMs is another plausible option that would be interesting to explore. We will try to incorporate it the the future set of experiments, towards the final version of the paper.

---

### Decision · Program_Chairs · 2019-12-19

**Decision:**

Accept (Poster)

**Comment:**

This work proposes use of two pre-trained FST models to explicitly incorporate semantic and strategic/tactic information from dialog history into non-collaborative (negotiation) dialog systems. Experiments on two datasets from prior work show the advantage of this model in automated and human evaluation. While all reviewers found the work interesting, they made many suggestions regarding the presentation. Author'(s) rebuttal included explanations and changes to the presentation. Hence, I suggest acceptance as a poster presentation.